# Airborne Radio-Echo Sounding Data Denoising Using Particle Swarm Optimization and Multivariate Variational Mode Decomposition

Yuhan Chen [1,2], Sixin Liu [1], Kun Luo [3], Lijuan Wang [2,4] and Xueyuan Tang [2,5,*]

1   College of Geo-Exploration Science and Technology, Jilin University, No. 938 Ximinzhu Street, Changchun 130026, China; yuhanc21@mails.jlu.edu.cn (Y.C.); liusixin@jlu.edu.cn (S.L.)
2   Key Laboratory of Polar Science of Ministry of Natural Resources (MNR), Polar Research Institute of China, Shanghai 200136, China; lijuan_wang@tongji.edu.cn
3   The Institute for Interdisciplinary and Innovate Research, Xi'an University of Architecture and Technology, Xi'an 710055, China; luokun@xauat.edu.cn
4   College of Surveying and Geo-Informatics, Tongji University, No. 1239 Siping Road, Shanghai 200092, China
5   School of Oceanography, Shanghai Jiao Tong University, Shanghai 200030, China
*   Correspondence: tangxueyuan@pric.org.cn

**Abstract:** Radio-echo sounding (RES) is widely used for polar ice sheet detection due to its wide coverage and high efficiency. The multivariate variational mode decomposition (MVMD) algorithm for the processing of RES data is an improvement to the variational mode decomposition (VMD) algorithm. It processes data encompassing multiple channels. Determining the most effective component combination of the penalty parameter ($\alpha$) and the number of intrinsic mode functions (IMFs) ($K$) is fundamental and affects the decomposition results. $\alpha$ and $K$ in traditional MVMD are provided by subjective experience. We integrated the particle swarm optimization (PSO) algorithm to iteratively optimize these parameters—specifically, $\alpha$ and $K$—with high precision. This was then combined with the four quantitative parameters: energy entropy, signal-to-noise ratio (SNR), peak signal-to-noise ratio (PSNR), and root-mean-square error (RMSE). The RES signal decomposition results were judged, and the most effective component combination for noise suppression was selected. We processed the airborne RES data from the East Antarctic ice sheet using the combined PSO–MVMD method. The results confirmed the quality of the proposed method in attenuating the RES signal noise, enhancing the weak signal of the ice base, and improving the SNR. This combined PSO–MVMD method may help to enhance weak signals in deeper parts of ice sheets and may be an effective tool for RES data interpretation.

**Keywords:** radio-echo sounding (RES); multivariate variational mode decomposition (MVMD); intrinsic mode functions (IMFs); particle swarm optimization (PSO); East Antarctic ice sheet

## 1. Introduction

Rapid climate change has led to a significant loss of mass of the Antarctic ice sheet (AIS) [1]. Research has revealed that the East Antarctic ice sheet may be more vulnerable than expected; parts of it may be as unstable as parts of the West Antarctic ice sheet [2–4]. Determining the stability of different regions of the AIS is crucial. An important factor limiting the ability of ice sheet models to understand and predict future ice sheet behavior is a lack of understanding of the englacial structure of the AIS [5,6]. A radio-echo sounding (RES) survey is an effective way to quantify the englacial information of the AIS, as well as the basal conditions [7,8]. RES is a tool with a high penetration depth that can be used to obtain high-resolution ice layer information. RES has been widely used to obtain the geometric characteristics of the ice bed, the ice internal layer structure of the ice sheet, information regarding the past subglacial landscape beneath the AIS, ice dynamics,

and to locate old ice cores [9–16]. Although significant progress has been achieved in the application of ice radar in Antarctica, substantial limitations remain. These include widespread noise interference and signal attenuation, which limit the characterization of ice layer structures and subglacial morphology features in radar images. This affects the accuracy of the feature extraction of ice sheets [17,18].

The attenuation of RES signals is mainly caused by the dielectric constant and conductivity of the ice [19]. The electrical conductivity of ice is affected by the content of impurities, intra-ice ions, and gas bubbles produced by the deposition of sea salt and acids emitted into the atmosphere by volcanic eruptions [15,20–22]. The dielectric constant is related to ice chemistry variations, ice fabrics, and the pressure and temperature of ice [23,24]. All these factors result in the corresponding attenuation of electromagnetic waves propagating within the ice. Airborne RES is usually emitted in the form of a pulse modulation. The geometric diffusion of the radar wave also causes attenuation [25]. The interaction between an aircraft and the antenna, the thermal noise of the system, and the clutter all produce noise interference [26–28]. These factors can result in a lack of valid information in the radar-reflected echo, which reduces the signal quality. Thus, the geological information cannot be reconstructed and restored. Identifying and eliminating noise and clutter, as well as improving the extraction accuracy of geometric features, are important aspects of RES signal processing. Many denoising methods based on the characteristics of radar detection have been proposed; these focus on radar signal principles such as filtering, wavelet transform, Fourier transform, low-rank approximation, and deep learning [29–33]. Fourier transform and wavelet analysis are suitable for the decomposition of linear stationary signals. The basis of decomposition is limited by the time–frequency resolution, which cannot be fully applied to non-linear, non-stationary signals [30]. The low-rank approximation is essentially single-channel denoising; it does not effectively use the inter-channel coherence of the signal [34]. Methods such as convolutional neural networks (CNNs), which denoise signals using deep learning, rely on a large number of labeled training samples. This limits their application [33]. As the pursuit of novel signal processing methods gains momentum, such issues require urgent attention. Traditional approaches are inadequate for contemporary demands.

We theorized that the decomposition of airborne RES data into intrinsic mode function (IMF) components may be a more effective method to process stochastic non-smooth signals [35]. In 1998, the empirical mode decomposition (EMD) method was established. This decomposes a signal into multiple IMFs of different frequencies and applies the Hilbert transform (N. E. Huang). Parameters such as an instantaneous frequency and an instantaneous phase can be obtained through EMD processing; thus, this method has a high adaptability and time–frequency resolution. It is mainly used to process non-linear, non-stationary signals, which are useful in navigation, meteorology, seismic record analyses, ground-penetrating radar data processing, mechanical fault diagnosis, and biomedical applications [36–44]. The EMD method has been improved over time, but this decomposition method continues to encounter challenges, such as the mathematical model not being rigorous enough and noise sensitivity [45]. In 2014, Dragomiretskiy et al. introduced the variational mode decomposition (VMD) algorithm [46]. VMD decomposes the signal into non-recursive variational modes; it possesses improved noise robustness. Through the iterative pursuit of optimal solutions within the variational model, each signal component becomes stationary, solving the mode-mixing phenomenon. However, it lacks self-adaptability; the degree of freedom of the parameters is also significant. The stripe problem also remains. In 2019, Rehman et al. introduced multivariate variational mode decomposition (MVMD) [47], an adaptive frequency–domain division method that decomposes several IMFs from a signal. MVMD extends the variational model of VMD to process multivariate data. The algorithmic process is based on common frequency components across all channels to construct a variational optimization problem. It extracts a multivariate modulation oscillation finite bandwidth mode set. The center frequency and bandwidth of the IMF can then be determined by the optimal set, which is then searched by the variational

model. MVMD has the advantages of VMD while extending it from a single channel to a multichannel with good modal alignment. This method avoids both the stripe problem of VMD and the drawbacks of multivariate empirical mode decomposition (MEMD) [47,48]. It possesses the obvious advantage of a high resolution in the time–frequency analysis of data. MVMD shares a parameter reliance with VMD; this requires manual input for the penalty parameter ($\alpha$) and the number of IMFs (*K*).

To overcome the problem that MVMD needs to rely on manual experience to select parameters, we propose an MVMD denoising method that incorporates particle swarm optimization (PSO) [49–56]. As a global search procedure, PSO has the advantages of higher accuracy, higher operability, fewer parameter settings, and faster convergence; it has been widely used in engineering and geophysical fields [57]. First, the airborne RES data are read, and the PSO algorithm determines the optimal parameter combinations for MVMD. Next, by computing the energy entropy of the decomposed IMFs, ineffective components are removed. Finally, the effective components are optimally combined according to the signal-to-noise ratio (SNR), peak signal-to-noise ratio (PSNR), and root-mean-square error (RMSE) and are then accumulated to obtain the denoised reconstructed radar signal, thus achieving a denoising effect. The contributions of this method are as follows: (1) it provides a new MVMD method for airborne RES data processing of the East Antarctic ice sheet, which realizes the sequential decomposition of the signal from a low frequency to a high frequency; (2) combining MVMD with the PSO algorithm (PSO–MVMD) solves the problem of the reliance of MVMD on manual experience to set the parameters; (3) it produces a reliable denoising effect that enables a focus on deep glaciers and reveals high-resolution subglacial basement structures, improves the extraction accuracy of the geometric features, and obtains accurate geological boundary conditions in Antarctica.

## 2. Materials and Methods

### 2.1. Principles of MVMD

Compared with VMD, MVMD serves as an extension. It transitions multivariate data from a one-dimensional space to multidimensional spaces, which is convenient when using multivariate or multichannel signals. It ensures the consistency of the frequency of multichannel decomposition. The decomposition steps are as follows. First, a multivariate constrained variational model is constructed. This results in the emergence of an associated optimization problem, as articulated in Equation (1). Second, the multivariate constrained variational model is solved; this adaptively decomposes the frequency bands of the signals by updating the relationship to obtain the *K* of the IMFs [47].

For the input multivariate signal $x(t)$, MVMD can extract the *K* multivariate modulation oscillation signals $U_k(t)$ from $x(t)$, which contains C channel data. This is expressed as

$$x(t) = \sum_{k=1}^{K} U_k(t) \tag{1}$$

where $U_k(t) = [U_1(t), U_2(t), \dots, U_c(t)]$.

MVMD extracts the set $\{U_k(t)|k = 1, 2, \dots, K\}$ of multimodulated oscillation signals in the input data. The total bandwidths of all modes are minimized. The raw signal can then be accurately reconstructed. The L2 norm of the gradient function of its analytic vector $U_+^k$ is used to estimate the bandwidth of $U_k(t)$. The L2 norm is defined as the open square of the sum of the squares of all elements of the vector. The multivariate function in MVMD that requires optimization is

$$f = \sum_k \left\| \partial_t \left[ e^{-j\omega_k t} U_+^k(t) \right] \right\|_2^2 \tag{2}$$

where $\omega_k$ is the center frequency of the extracted mode, which is a single-frequency component for the harmonic mixing of all $U_+^k(t)$. The bandwidth of the modulated multivariate

oscillation is estimated in two parts: one shifts the unilateral frequency spectrum of each channel of $U_+^k(t)$ by $\omega_k$ and the other takes the Frobenius norm of the matrix. Thus, $f$ is

$$f = \sum_k \sum_c ||\partial_t \left[ U_+^{k,c}(t) e^{-j\omega_k t} \right] ||_2^2 \tag{3}$$

where $U_+^{k,c}$ is the mode in channel $c$, and $k$ is the analytic modulated signal. The constrained optimization problem of MVMD is then

$$\begin{cases} \min\limits_{\{u_{k,c'}(t)\},\{\omega_t\}} \sum_k \sum_c \left\| \partial_t \left[ U_+^{k,c}(t) e^{-j\omega_k t} \right] \right\|_2^2 \\ \sum_k U_{k,c}(t) = x_t(t) c = 1, 2, \cdots, C \end{cases} \tag{4}$$

where $\{U_{k,c}(t)\}$ is an ensemble of multivariate modulated oscillations in channel $c$ and $\{\omega_k\}$ is the ensemble of the center frequencies of $\{U_{k,c}(t)\}$. The corresponding augmented Lagrangian function is

$$\begin{aligned} L(\{U_{k,c}(t), \{\omega_k\}, \lambda_{(c)}(t) = \alpha \sum_k \sum_c \left\| \partial_t \left[ e^{-j\omega_k t} U_+^{k,c}(t) \right] \right\|_2^2 + \| x_t(t) - \sum_k U_{k,c}(t) \|_2^2 \\ + \sum_c \langle \lambda_c(t), x_c(t) - \sum_k U_{k,c}(t) \rangle \end{aligned} \tag{5}$$

where $\alpha$ is the penalty factor, $\lambda_c(t)$ is the Lagrange multiplier, and $< >$ is the inner product.

In contrast to VMD, MVMD demonstrates a pronounced mode alignment property. This is particularly advantageous when dealing with multivariate signals. The mode alignment property of multivariate signals signifies that the signal components of a multivariate signal that are in the same frequency band as the modal component can be simultaneously decomposed into that modal. This is extremely important for the parallel analysis of multivariate data, especially in geophysical signal processing, where the decomposed mode components must be applied to enable the signal reconstruction. If there are oscillating signals in different frequency bands on the same mode component, the extracted multivariate signal features lose their significance if there are oscillating signals in different frequency bands on the same mode component.

As an illustrative example, we considered an analysis of a dual-channel composite input signal, as shown in Equations (6) and (7). In Figure 1, channel 1 is represented by the blue curve and encompasses signal components of 80 Hz and 40 Hz; channel 2, shown in red, consists of signal components at 80 Hz and 120 Hz. In decomposing the synthesized signal, the number of IMFs of *K* was uniformly set to 3 for both algorithms. Figure 2 reveals the VMD decomposition results; the 40 Hz component in channel 1 and the 80 Hz component in channel 2 were decomposed to IMF1 (Figure 2a,d). The same problem existed for IMF2. In the MVMD decomposition results (Figure 3), the common component of 80 Hz in both channels was extracted to IMF1, which verified that the MVMD method was equivalent to VMD in multivariate signal processing for the superiority of its mode alignment property.

$$a_1 = 1.2 \cdot cos(2\pi \cdot 80 \cdot t) + 0.8 \cdot cos(2\pi \cdot 40 \cdot t) \tag{6}$$

$$a_2 = 1.2 \cdot cos(2\pi \cdot 80 \cdot t) + 0.8 \cdot cos(2\pi \cdot 120 \cdot t) \tag{7}$$

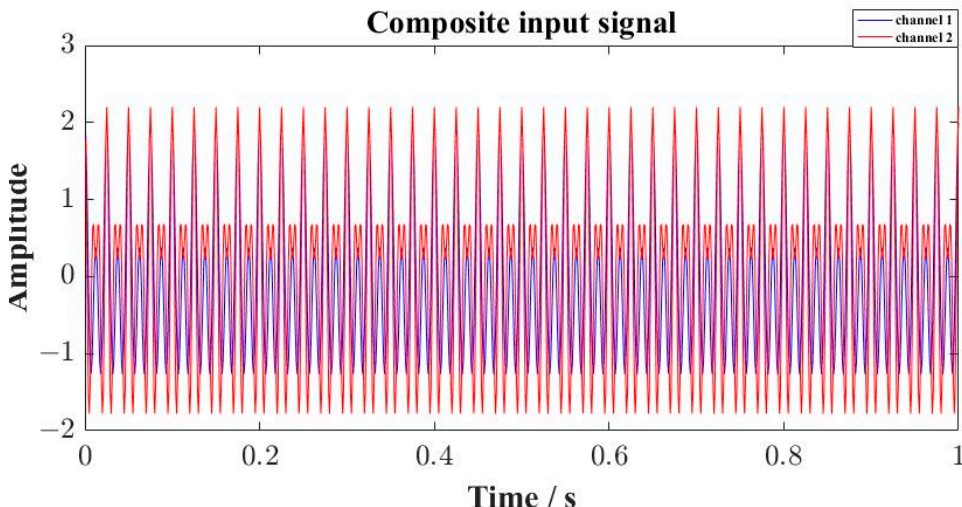

**Figure 1.** Dual-channel composite input signal.

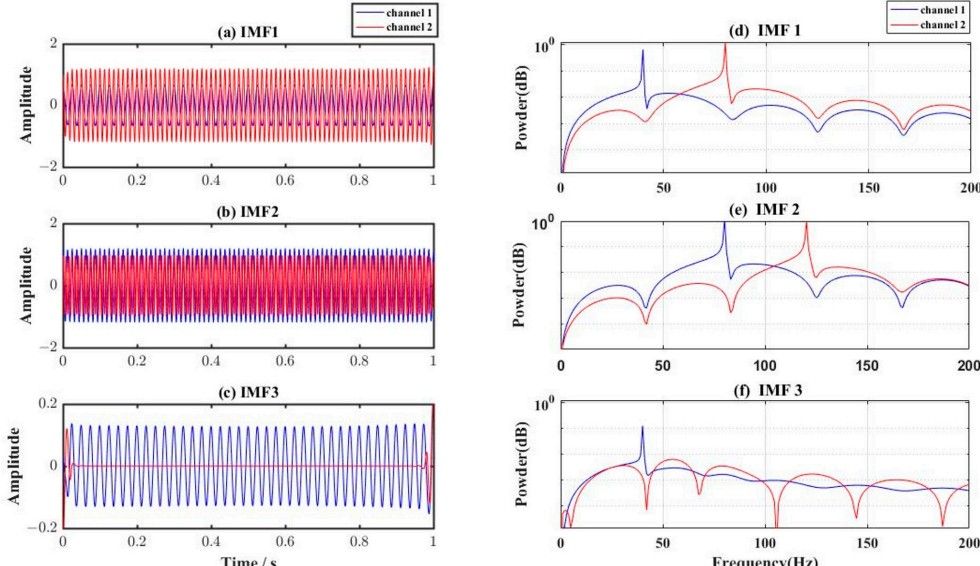

**Figure 2.** Decomposed IMFs using VMD for the dual-channel composite input signal ((**a**–**c**): waveform diagrams; (**d**–**f**): spectrum diagrams).

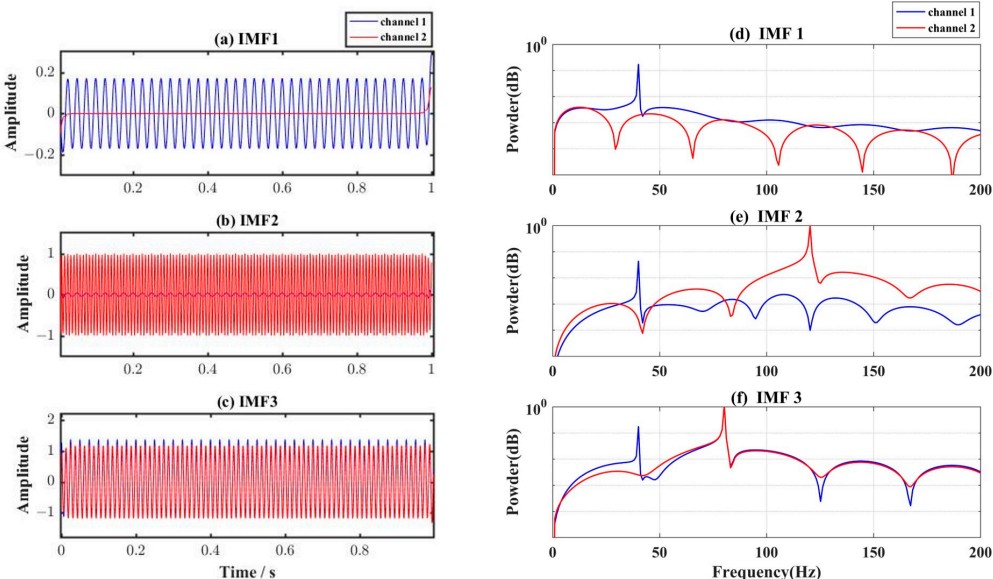

**Figure 3.** Decomposed IMFs using MVMD for the dual-channel composite input signal ((**a**–**c**): waveform diagrams; (**d**–**f**): spectrum diagrams).

### 2.2. The PSO–MVMD Algorithm for Optimal Parameters of α

The number of IMFs, *K*, and the penalty factor, *α*, are required to be set by virtue of manual experience when the MVMD algorithm decomposes the signal. *K* affects the decomposition effect of the signal; a large *K* over-decomposes the mode, whereas a small *K* may result in under-decomposition. The penalty factor *α* affects the IMF bandwidth and the speed of convergence; larger *α* values correspond with narrower bandwidths of IMFs [58]. We introduced the PSO algorithm to automate the optimization of *K* and *α*.

The PSO algorithm, which was proposed by Kennedy et al. in 1995 [49], simulates the unconscious foraging behavior of a flock of birds for its own state optimization. Each particle represents a bird, and the optimal solution equates to a food source. The particle swarm determines the distance and direction of the flight according to its individual experience and neighbor experiences. It updates the two attributes of the particle—position, and velocity—and continuously iterates to discover the optimal solution, thereby completing the search process.

If the number of particles is n, D-dimensional space can be searched by composing the set $X = \{X_1, X_2, \cdots X_n\}$. The position of the *i*-th particle in the solution space in the D-dimensional space is as follows (Equation (8)):

$$X_I = \{X_{i1}, X_{i2} \cdots X_{id}\} \tag{8}$$

The velocity of the *i*-th particle in the D-dimensional space is as follows (Equation (9)):

$$V_I = \{V_{i1}, V_{i2} \cdots V_{id}\} \tag{9}$$

The optimal position of the *i*-th particle, which is searched in the D-dimensional space (i.e., the best fitness value of the particle), is shown in Equation (10):

$$P_{best} = \{P_{i1}, P_{i2} \cdots P_{id}\} \tag{10}$$

The optimal position that the particle population historically searches for in D-dimensional space (i.e., the best fitness value of the population) is shown in Equation (11):

$$g_{best} = \{g_{i1}, g_{i2} \cdots g_{id}\} \tag{11}$$

The particle swarm updates the position and velocity through the best fitness value of the particle and the population. It continuously iterates to discover the optimal problem solution. $k$ is the number of current iterations; this was set to 10 in this paper. At the $(k+1)$th generation, the velocity and position of the particle are updated using Equations (12) and (13), respectively:

$$V_{id}^{k+1} = \omega V_{id}^k + C_1 r_1 \left( P_{id}^k - X_{id}^k \right) + C_2 r_2 \left( g_{id}^k - X_{id}^k \right) \tag{12}$$

$$X_{id}^{k+1} = X_{id}^k + V_{id}^k \tag{13}$$

where $i = 1, 2, \cdots, n$; $d = 1, 2, \cdots, D$; and $\omega$ is the inertia weighting factor, which determines the amount that the velocity is preserved in the last iteration. This is used to regulate the search range of the solution space. If the inertia weight value is large, the algorithm has a strong global search ability. It was set to 1.5 in this paper. $C_1$ and $C_2$ are acceleration constants, which determine the size of the amount of learning by the particle for the optimal position and control the search step for the individual and the population, respectively. This is tested by a benchmark function set; the optimal interval for the acceleration constant ranges from 0.5 to 2.5 [59]. In this paper, it was set to 1.5 and 1.0. $r_1$ and $r_2$ are random numbers within [0, 1]. $P_{iD}^k$, $X_{iD}^k$, $g_{iD}^k$, and $V_{iD}^k$ are generated in the $k$th iteration.

The envelope entropy $E_p$ is the fitness function; the envelope entropy $E_p$ of the envelope signal $a(j)$ is then obtained after signal $x(j)(j = 1, 2, \cdots, N)$ through Hilbert demodulation. This is calculated as follows:

$$\begin{cases} E_p = -\sum_{j=1}^N p_j \log_{10} p_j \\ p_j = \frac{a(j)}{\sum_{j=1}^N a(j)} \end{cases} \tag{14}$$

where $a(j)$ is the envelope signal, $p_j$ is the normalized form of $a(j)$, and $E_p$ is used to quantitatively describe the sparsity of the raw signal. A smaller value of $E_p$ represents sparser signals (i.e., less noise).

At this time, the process of selecting a group of $[k, \alpha]$ as the parameters for the MVMD algorithm by the PSO algorithm is that the signal is decomposed by MVMD at a certain particle position to obtain $K$ IMFs. The $E_p$ value of each IMF is calculated to discover its minimum value as the local minimal entropy value (i.e., if the fitness function is $E_p$, the search target is the local minimal entropy value). The optimal parameter combination $[k, \alpha]$ search flowchart of the MVMD algorithm is presented in Figure 4.

To test the combined PSO–MVMD method, we used a composite input signal composed of a 50 Hz sine wave, cosine waves ranging from 0.1 to 0.2 s, 0.4 to 0.5 s, and 0.7 to 0.8 s, and a 0.5 to 1.0 s cosine wave along with Gaussian white noise. The composite input signals of the time–domain and frequency–domain waveforms of this example are shown in Figure 5. Its constituent frequencies were 50 Hz, 100 Hz, and 250 Hz, respectively. In the process of particle swarm optimization, the local minimal entropy value varies with the evolution algebra of the population. Figure 6a illustrates the distribution of the particle swarm following evolution; each solid circle is the distribution of a particle, and the color indicates its corresponding entropy value. Figure 6b reveals that the 5th generation demonstrated a local minimal entropy value of 6.417. The best parameter combination derived from this optimization process [9, 2761] is displayed in Figure 6c; the circled red portion is the ultimate selected point (i.e., K = 9 and $\alpha$ = 2761). PSO–MVMD was performed on the noisy composite input signal to obtain nine IMFs. Figure 7a–i are waveform diagrams; Figure 7j–r are spectrum diagrams. The sine wave of 50 Hz was mainly decomposed into IMF2; the cosine waves of 0.1–0.2 s, 0.4–0.5 s, and 0.7–0.8 s of 100 Hz were mainly decomposed into IMF3; the cosine wave of 0.5–1.0 s of 250 Hz was mainly decomposed into IMF5; and the other IMFs were noise components. We observed that IMF2 was in good agreement with $x_1$. The IMF3 frequency was in the range of 96–106 Hz for 0.1–0.2 s, 0.4–0.5 s, and 0.7–0.8 s.

There were a few small disparities compared with $x_2$. IMF5 was basically in agreement with the $x_3$ frequency. This was an acceptable result for MVMD [47]. PSO–MVMD sequentially decomposed the signal from low to high frequencies and achieved an effective separation of the components with a high time–frequency resolution and accuracy, thus proving the effectiveness of the method.

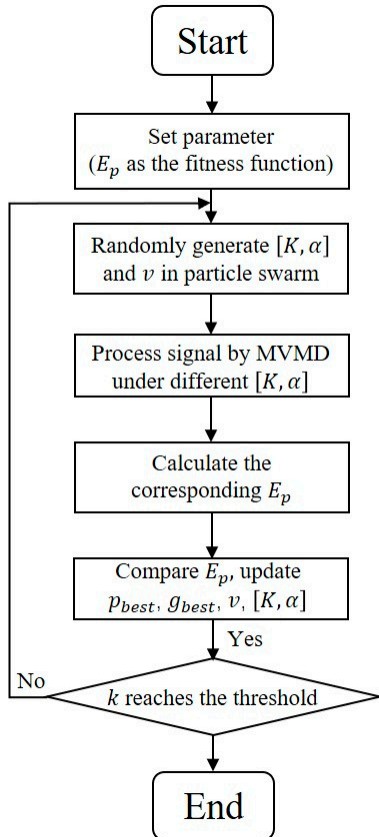

**Figure 4.** Flowchart of PSO algorithm searching for $[k, \alpha]$.

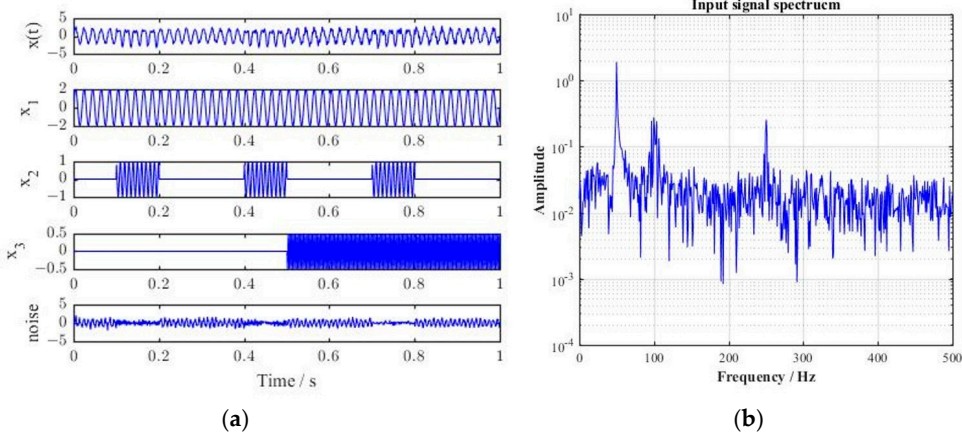

**Figure 5.** (**a**) Composite input signal and its component signals; (**b**) spectrum of the composite input signal.

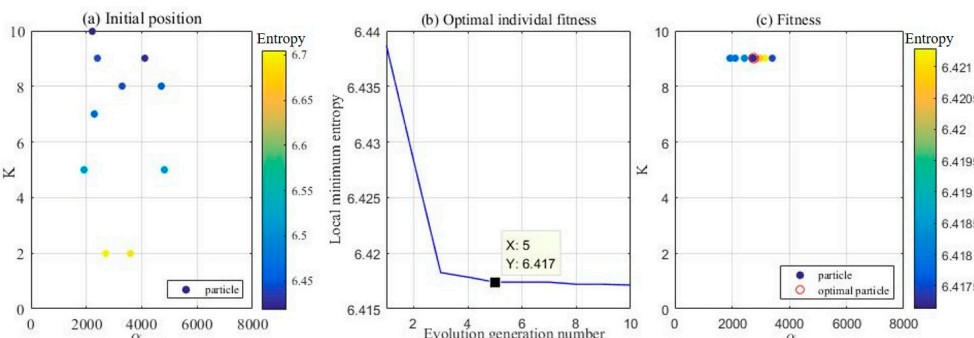

**Figure 6.** (**a**) Initial position of particles for component signals; (**b**) local minimal entropy value with PSO evolution; (**c**) final position of the particle.

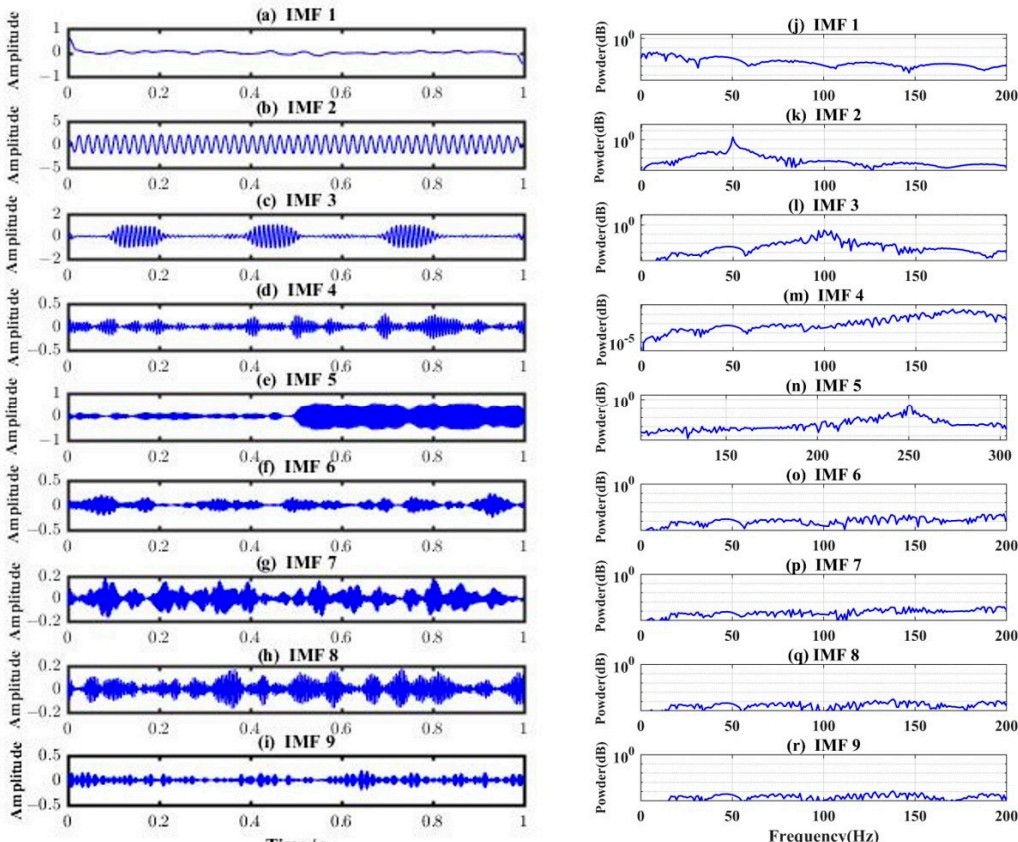

**Figure 7.** Decomposed IMFs using MVMD for the composite input signal ((**a**–**i**): waveform diagrams; (**j**–**r**): spectrum diagrams).

### 2.3. Energy Entropy Used to Select the Effective IMF

In this paper, we computed the energy entropy to efficiently select the IMFs that contained useful information [56]. We defined the energy of the *i* IMF as $E_i$. The ratio between the energy of the i IMF and the total energy is as follows:

$$P_i = E_i / \sum_{i=1}^{k} E_i \tag{15}$$

The energy entropy of the i IMF is

$$H_i = -P_i \log P_i \tag{16}$$

Energy entropy values are a valuable tool to quantify the regularity of IMFs. The relationship is straightforward; higher entropy values indicate more dispersed energy distributions, signaling greater instability and increased signal complexity. The higher the energy entropy, the more noise is contained in the signal.

### 2.4. The SNR, PSNR, and RMSE for the Evaluation of the Denoising Effect

Three parameters, SNR, PSNR, and RMSE, were used to measure the denoising effect of the method proposed in this paper before and after signal denoising to evaluate the degree of noise reduction. The SNR, PSNR, and RMSE are calculated as follows:

$$SNR = 10\log_{10}\left(\frac{\sum_{n=1}^{N} a(n)^2}{\sum_{n=1}^{N}(a(n) - b(n))^2}\right) \tag{17}$$

$$PSNR = 10\log_{10}\left(\frac{\max(b(n))^2}{\frac{1}{N}\sum_{n=1}^{N}(a(n) - b(n))^2}\right) \tag{18}$$

$$RMSE = \sqrt{\frac{1}{N}\sum_{n=1}^{N}(a(n) - b(n))^2} \tag{19}$$

where $a(n)$ is a raw signal with noise, $b(n)$ is the denoising signal, and $N$ is the number of samples. An elevated SNR signifies a greater presence of informative signal contents within the overall signal, underscoring an enhanced noise reduction outcome. The PSNR is an objective evaluation metric used to evaluate noise levels. An elevated PSNR indicates more effective noise reduction. The RMSE serves as a gauge to measure the extent of the deviation between the signal and the mean value of the initial data after noise reduction. A smaller RMSE indicates more effective noise reduction.

## 3. Results

### 3.1. Radio-Echo Sounding Data

The airborne RES data used were survey lines that were collected by the 32nd Chinese National Antarctic Research Expedition (CHINARE 32) using the Snow Eagle 601 aero-geophysical platform (The Snow Eagle 601 is a modified DC-3 aircraft by Basler Turbo Conversions in Wisconsin, USA and Lake Central Air Services in Ontario, Canada.) to perform aerial surveys. This dataset was obtained from 19 geophysical survey flights from 15 December 2015 to 29 January 2016 in Princess Elizabeth Land (PEL) of East Antarctica, which covered 866,000 square kilometers. The system was a phase-coherent RES system that was similar to the High-Capability Airborne Radar System (HiCARS) [60]. The central frequency of the antenna was 60 MHz, and the sampling frequency was 50 MHz.

The East Antarctic RES data lines used in this paper were located between 78°S~79°S and 74°E~75°E (X: 1000~1300 km; Y: −200~220 km). Figure 8 presents the location of the survey line in Antarctica. The background image in Figure 8 is the Bedmap2 elevation [61]. The data used in this paper are 6334 traces, with 3200 samples in each trace. The average track space was approximately 20 m. This data was, therefore, a two-dimensional matrix of $3200 \times 6334$. The data format was a binary mat file compiled using MATLAB (2021b). The data size was 53.9 MB.

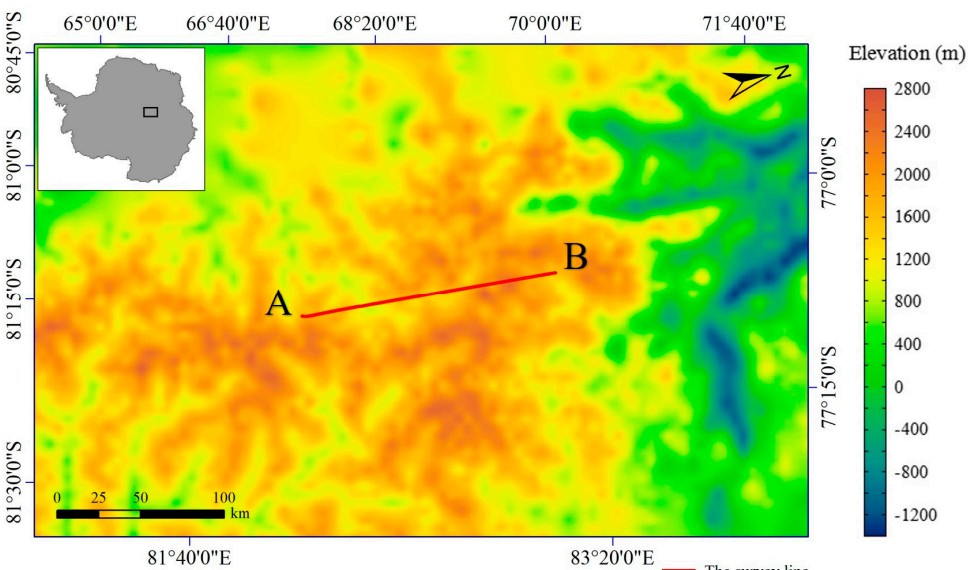

**Figure 8.** Location of the survey line AB in the Antarctic (background image is the Bedmap2 elevation).

The raw data were processed using down-conversion, DC offset removal, pulse compression, filtering, coherent stacking (10 times), and non-coherent stacking (5 times) to produce a field data product, "pik1", which is easy to use in other applications [62]. As shown in Figure 9, a clear ice surface and bedrock cross-section could be seen. The denoising process was applied to the airborne RES data.

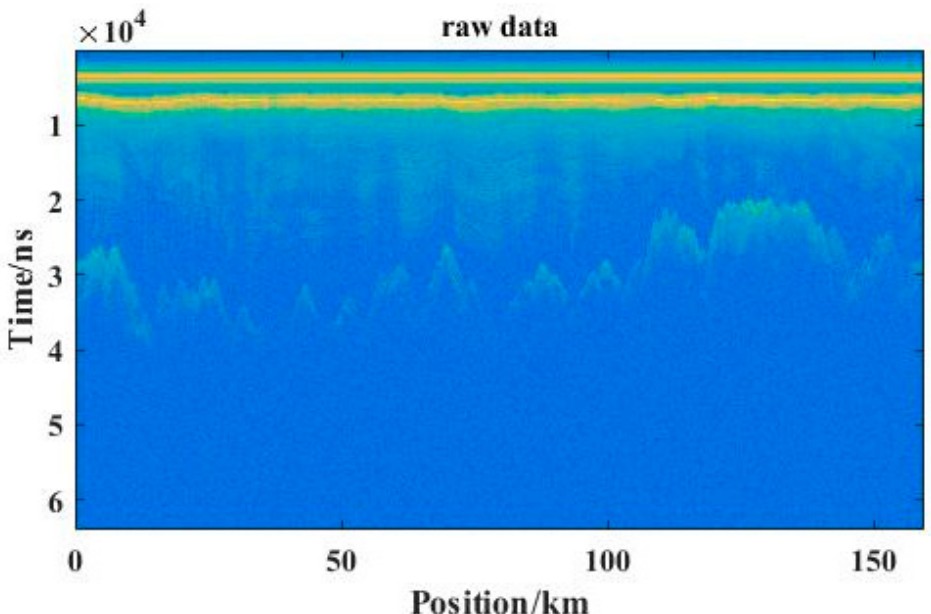

**Figure 9.** Raw data profile in Figure 8.

### 3.2. Data-Processing Procedure

On the basis of the above research, we proposed a PSO–MVMD denoising method. This method entailed the optimization of parameters and the strategic selection of IMFs for an efficient reconstruction. The general framework is presented in Figure 10. This method was applied to the RES data.

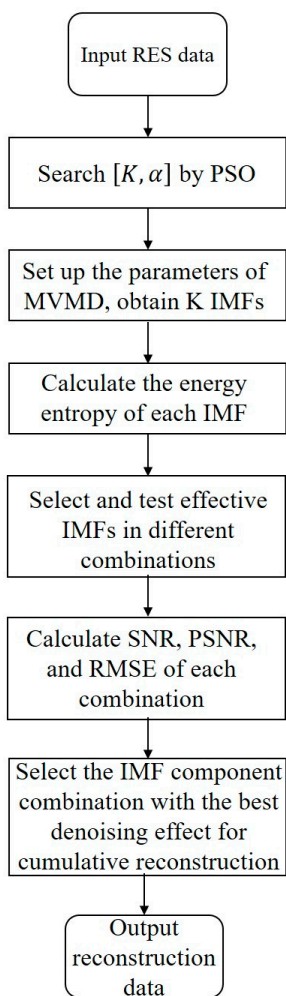

**Figure 10.** Flowchart of data-processing procedure.

### 3.3. Parameter Settings

Figure 11a presents the distribution of the particle swarm following evolution. Each solid circle signifies the distribution of an individual particle, and colors represent the entropy values. In Figure 11b, the evolution of the local minimal entropy values with population evolution is depicted during the particle swarm optimization process. Figure 11b highlights that the 4th generation yielded a local minimal entropy value of 8.487. The optimal parameter combination obtained from the optimization is presented in Figure 11c; this was $[k, \alpha] = [7, \ 3336]$. The circled red portion is the ultimate selection point.

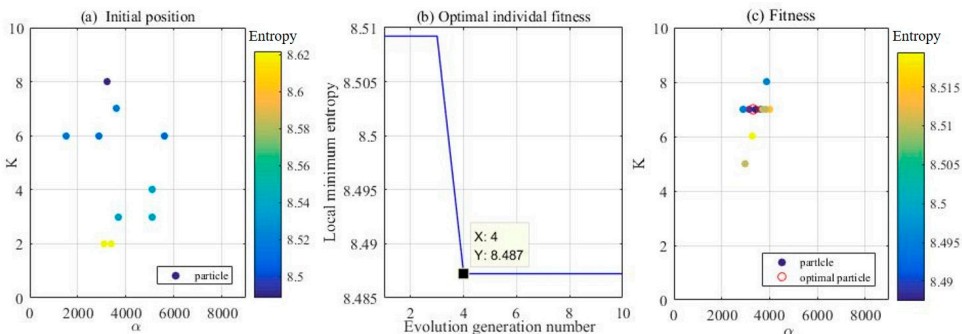

**Figure 11.** (**a**) Initial position of particles for RES data; (**b**) local minimal entropy value with PSO evolution; (**c**) final position of the particle.

### 3.4. Data Reconstruction

Figure 12 presents the seven IMFs obtained by MVMD decomposition. To quantitatively determine the effective IMF, the energy entropy of the seven IMFs was calculated, as shown in Table 1. IMF1, IMF2, and IMF4 had lower energy entropy values; therefore, we focused on different combinations of IMF1, IMF2, and IMF4. According to the combination results (Table 2), the combination of IMF1, IMF2, and IMF4 had the highest SNR of 77.89 and PSNR of 11.35, and the smallest RMSE of 2096.91. Figure 13 presents a reconstruction of the combination.

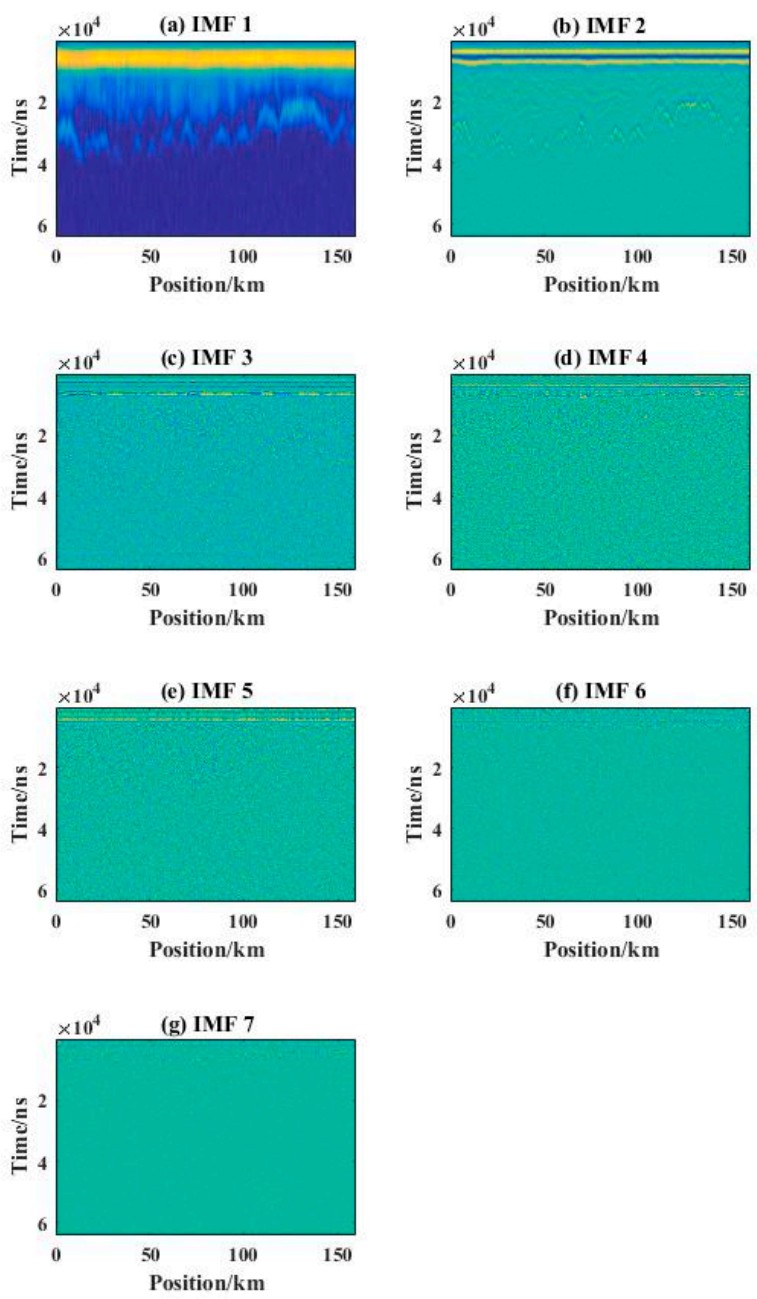

**Figure 12.** IMFs after MVMD decomposition.

**Table 1.** Energy entropy of each IMF.

| IMF | IMF1 | IMF2 | IMF3 | IMF4 | IMF5 | IMF6 | IMF7 |
|---|---|---|---|---|---|---|---|
| Energy entropy | 0.2777 | 0.2776 | 0.2784 | 0.2777 | 0.2780 | 0.2781 | 0.2784 |

**Table 2.** IMF1, IMF2, and IMF4 combination test results.

| No. | IMFs | SNR | RMSE | PSNR | Note |
|---|---|---|---|---|---|
| 1 | 1, 2, and 4 | 77.89 | 2096.91 | 11.35 | Selected |
| 2 | 2 and 4 | 0.05 | 102,490.64 | 10.21 | |
| 3 | 1 and 2 | 73.80 | 2571.34 | 11.08 | |
| 4 | 1 and 4 | 56.80 | 6020.48 | 10.24 | |

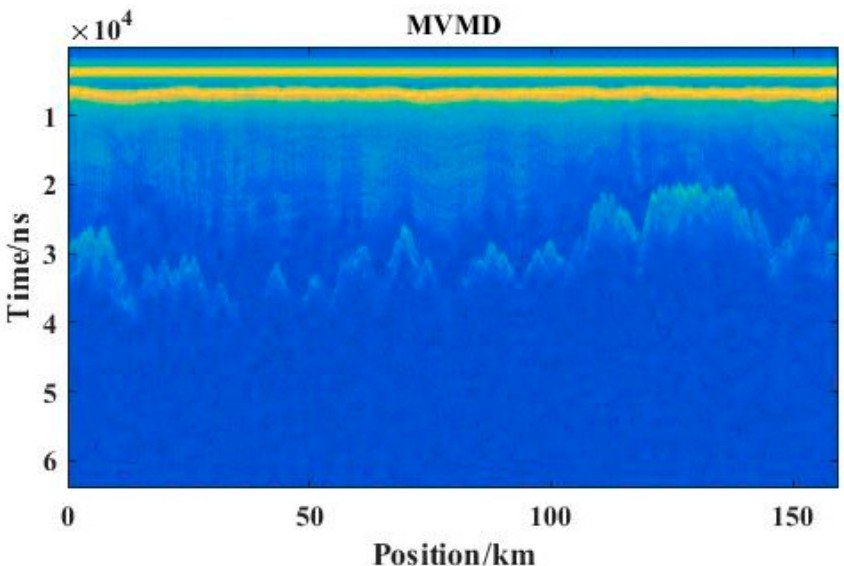

**Figure 13.** MVMD reconstruction profile.

## 4. Discussion

Figure 12 demonstrates that IMF1 and IMF2 contained the main bedrock interface. IMF1 and IMF2 contained a large amount of internal reflection information from the ice layer, but the judgment based on manual experience was not rigorous enough. When comparing the energy entropy values of every component in Table 1, IMF2, IMF1, and IMF4 had lower energy entropy values and contained more useful information, whereas IMF7, IMF3, and IMF6 had higher energy entropy values and contained a significant amount of noise and less useful information. To confirm the accuracy of the energy entropy, we reconstructed the two combinations of IMF1, IMF2, and IMF4 (the combination with lower energy entropy) and IMF7, IMF3, and IMF6 (the combination with higher energy entropy), as shown in Figure 14. We calculated the RMSE, PSNR, and SNR (Table 3), which highlighted notable distinctions between the two ensembles. The combination of IMF7, IMF3, and IMF6 failed to effectively extract useful signals, and the internal reflection information of the ice layer could not be seen. The bedrock interface and ice surface could not be clearly seen, as presented in Figure 14a. In contrast, the IMF1, IMF2, and IMF4 combination clearly defined the bedrock interface, and the geological characteristics were preserved. When comparing the magnitude of the SNR, PSNR, and RMSE, we observed that the higher energy entropy combination had a very low SNR and PSNR and a very large RMSE, which retained a lot of noise and removed the effective signal. The SNR and PSNR of the higher energy entropy combination were high, and the RMSE was small, which proved the effectiveness of discriminating IMFs by energy entropy.

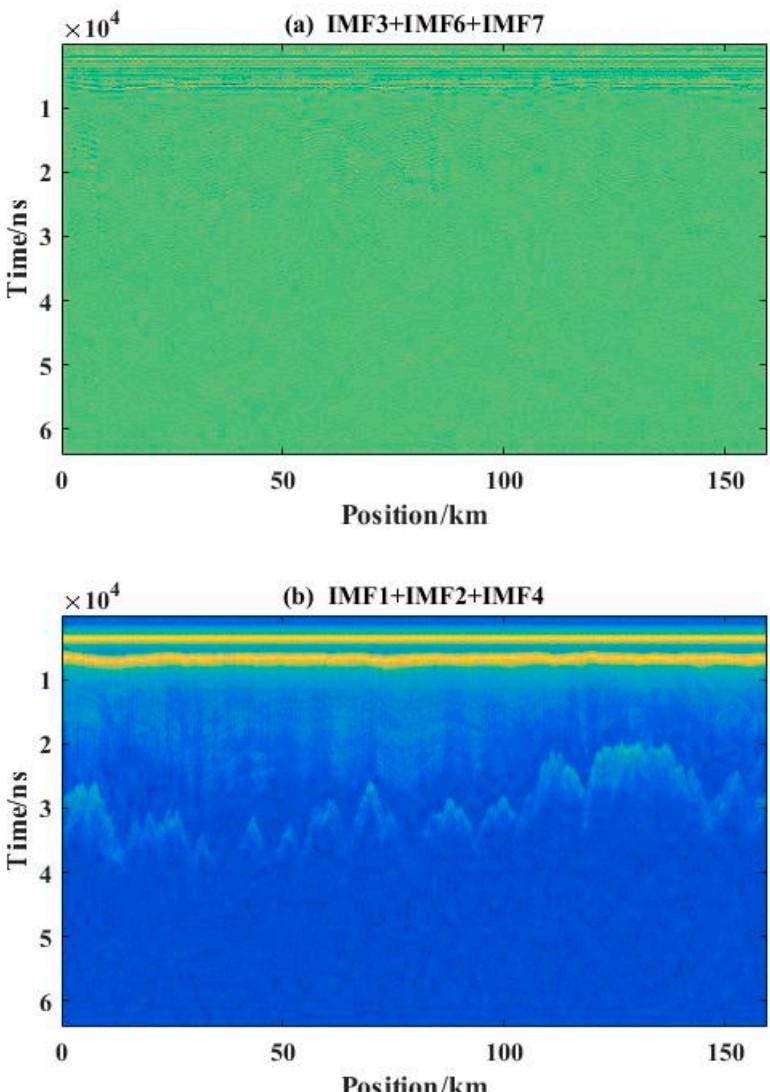

**Figure 14.** (**a**) IMF3 + IMF6 + IMF7 combination profile; (**b**) IMF1 + IMF2 + IMF4 combination profile.

**Table 3.** Test results of the energy entropy higher combinations and energy entropy lower combinations.

| Combination | SNR | RMSE | PSNR |
|---|---|---|---|
| IMF3, IMF6, and IMF7 | 0.01 | 102,720.86 | 9.16 |
| IMF1, IMF2, and IMF4 | 77.89 | 2096.91 | 11.35 |

As shown in Table 1, the differences in the energy entropy values of the IMFs were on the order of $10^{-5}$. The highest energy entropy value differed from the lowest energy entropy value by 0.0008. This was because the raw data already contained fewer noise signals and were of good quality. The raw data could already visualize the general orientation of the ice surface as well as the bedrock interface, and no obvious high-frequency noise was seen.

Based on the results listed in Table 2, we obtained a profile of the best combination reconstruction, as shown in Figure 13. For a clearer comparison with the raw data, we created a detailed figure (Figure 15). Figure 15a reveals the raw data profile, and Figure 15b reveals the profile after the PSO–MVMD reconstruction. A comparison between Figure 15a,b revealed three obvious effects. Red rectangular frame I indicates the enhanced continuity of the bedrock interface; there was a weak signal enhancement in the deep glacier, and the horizontal resolution was improved. A greater number of reflective layer surfaces in

the deep layers are identified in red rectangular frame II; the bedrock interface was more prominent and reflected hidden basement environment information. Red rectangular frame III demonstrates that a greater number of subglacial bedrock structures were revealed, including signals from the deep layers and the bed. The geometry of the bedrock interface was more accurate. The results indicated more complex geometrical characteristics of the subglacial basement topography, which helped to reveal further bedrock properties as well as basement sediments, subglacial water units, geological structures, etc. [63,64].

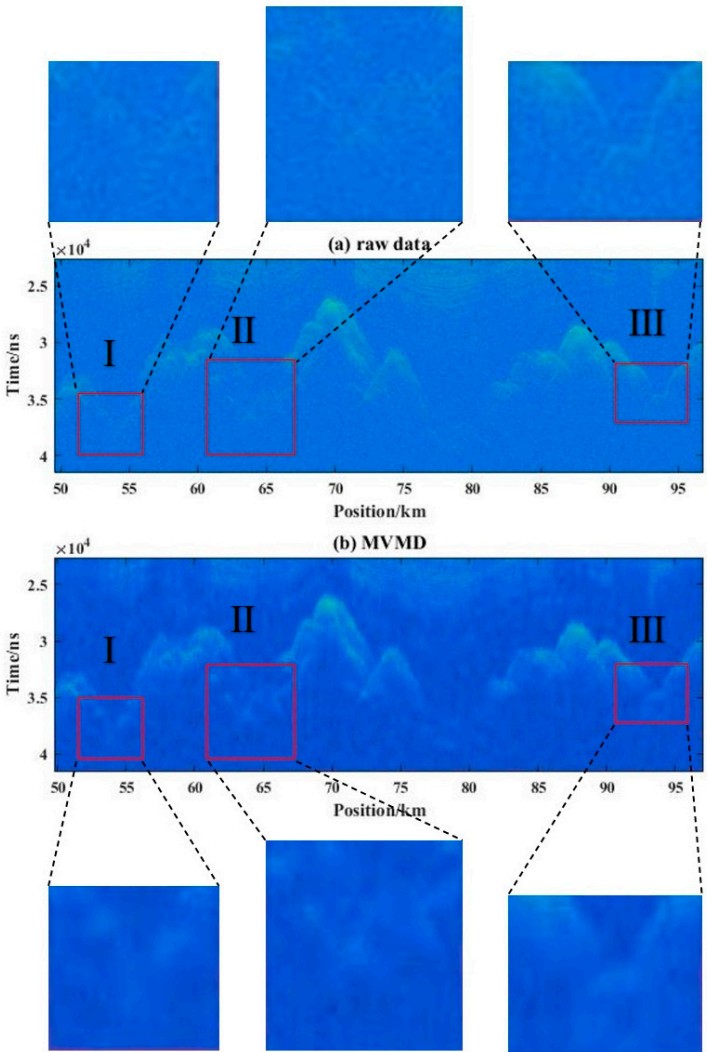

**Figure 15.** (**a**) Raw data detail profile; (**b**) MVMD reconstruction detail profile.

We used the same parameter settings for VMD as for MVMD; i.e., we decomposed the RES data into seven IMFs and compared the computational efficiencies of the two methods. MVMD ran for 3534.086 s and VMD ran for 9459.076 s (Figure 16). This was because VMD is a single-channel processing method, and MVMD is a multichannel processing method; in comparison, MVMD has a higher computational efficiency. The method produced a sequential signal decomposition from low to high frequencies. This could overcome the limitation of single-radar signal processing, presenting a high-efficiency, adaptive, and multichannel simultaneous-processing denoising method. MVMD has been applied in the field of geophysics, but the K value is usually selected by constant testing. In this paper, MVMD was combined with the PSO algorithm, which automatically selected K and $\alpha$. This resulted in a more intelligent denoising effect [65–69]. The fundamental objectives of airborne RES data processing are to suppress noise, accentuate effective signals, and accurately reflect the internal characteristics of an ice sheet. The combined PSO–MVMD

method proposed in this paper focused on the enhancement of weak signals in deep ice layers. This effectively improved the SNR and PSNR of the radar signals as well as the recognition accuracy of bedrock interfaces. This method requires refinement and advancement in the future to more effectively expose complex geographic environments under the ice by processing RES data and to provide a basis for the interpretation of ice sheet flows and the stability of the water circulation system under the ice.

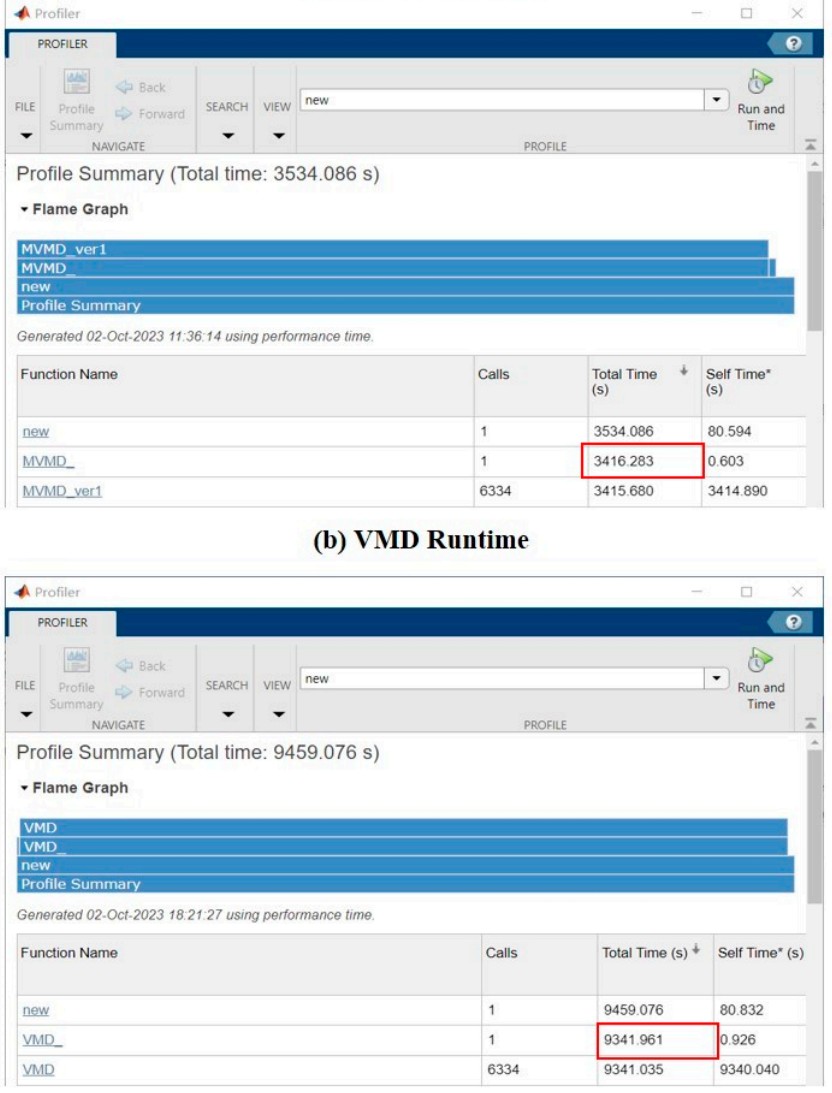

**Figure 16.** Decomposition of RES data into 7 IMF runtime comparisons: (**a**) MVMD runtime; (**b**) VMD runtime.

## 5. Conclusions

In this paper, we proposed a joint denoising method that combined PSO with MVMD. We selected the best combination of four parameters according to the energy entropy, SNR, PSNR, and RMSE. We applied this method to the denoising of RES signals, achieving an ideal denoising effect. As a powerful tool, MVMD can sequentially decompose signals in the frequency domain from low to high frequencies with very high time–frequency resolution and accuracy. Using multivariate signal processing, we demonstrated the superiority of MVMD over VMD for higher computational efficiency. Regarding the parameter setting of MVMD, the use of PSO realized an optimal automated search for the parameters of MVMD. This solved the problem of the two parameters, K and $\alpha$, requiring experience. The method

was accurate and effective. In the selection of IMFs, energy entropy was used to identify the effective IMFs. The best combination of reconstructed signals was then determined based on the SNR, PSNR, and RMSE. This reduced subjective judgments and improved the efficiency of data processing as well as the SNR. We applied this joint denoising method to an airborne RES data survey of the East Antarctic ice sheet; this method enhanced the continuity of the weak basal signal and the bedrock interface. The accuracy of the ice sheet structure and deep valley morphometric features was improved. In the future, we will enhance the quantitative treatment of RES data by improving the method. This method could be widely used to denoise RES data and restore hidden information in the bedrock interface, facilitating the further interpretation of ice dynamics.

**Author Contributions:** Conceptualization, X.T.; methodology, S.L. and Y.C.; software, Y.C., K.L. and L.W.; validation, Y.C. and X.T; data curation, X.T.; writing—original draft preparation, Y.C.; writing—review and editing, X.T. and S.L.; supervision, X.T. and S.L. All authors have read and agreed to the published version of the manuscript.

**Funding:** This study was supported by the National Natural Science Foundation of China (Nos. 42276257, 41876230, 41941006).

**Data Availability Statement:** The data presented in this study are available on request from the corresponding author. In future work, we will also upload data to https://datacenter.chinare.org.cn/data-center/dindex.

**Acknowledgments:** We express our appreciation to the Polar Research Institute of China for data support.

**Conflicts of Interest:** The authors declare no conflict of interest.

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
