# Peer review of "Airborne Radio-Echo Sounding Data Denoising Using Particle Swarm Optimization and Multivariate Variational Mode Decomposition"

_remotesensing, doi:10.3390/rs15205041_

Round 1

Reviewer 1 Report

The quality of the paper should be undoubtedly improved. 

Starting from technical aspect: references should be properly formatted, addressed (see third paragraph in page 2, where ref. [13] is mentioned and then story goes about some other ref. in next sentence?... use brackets and line for interval like [16 - 22], etc.). Carefully edit text for free spaces ("newmethod", same page, extra space before a comma in many words, space missing when starting a new sentence on several positions in text, etc.) and use of capital letters (when starting a new sentence in the middle of previous sentence, for example...) Lowercase and uppercase letters in u - terms after eq. (1) should be unified, "L2 norm" should be explained, "U+k" or "Uk+" labeling should be resolved and unified, in Fig. 1 left both channels are named "channel 1", in pages 6 and 7 equations should be addressed properly (not "as follows 8.. space 9... equation 10..." etc. Please use "eq. (8)" and accordingly.), terms "pj" and "p(j)" should be resolved and unified, in eq. (14) either "l" and "g" are explained or "log" (most probably?) is used, discussion in page 8 about results presented in Fig. 7 mention "IMF6" - while actually IMF5 is presented in Fig. (7)?, while in legend in the same figure subfigures are not addressed correctly.   

English should be improved, typing errors corrected.

More important: obviously smaller noise level in segments of time corresponding with x2 signal presence in Fig. 5a are not explained, nor discussed. Thresholds or any other aspect of detection corresponding with situation presented in Fig. 7 is not explained. How, for example, detection for IMF2, IMF3 and IMF5 can be implemented in reliable manner, without false detection of strong signal presented for IMF4 in the same figure? Within the two paragraphs in page 13 authors claim that Table 1 shows some IMF "have lower energy entropy" and later that "energy entropy values do not show great differences" - so, what is relevant here? Following presentation and discussion in Section 4 is unclear, not providing any comparison with existing methods. Thus, conclusion claiming "superiority of MVMD over VMD" does not stand - this is not proven in manuscript. 

All the aspects listed above should be improved before eventual consideration of publication. 

English should be improved, many typing and other technical errors corrected - please, see previous comment.

Reviewer 2 Report

My comments are:

1)      The algorithm proposed in this article is not validated with the state-of-the-art methods. Its quite surprising in scientific publishing. Compare your proposed algorithm with 4/5 recently published relevant literatures. You have only shown a comparison of Intrinsic Mode Function components.

2)     While comparing your results with state-of-the-art methods, show visual results of 6/7 images.

3)      References are not in a proper format. All the references should be uniform and follow the same style. Look the MDPI manual and write all the references in a uniform format. In [34], “2021” appeared twice. Correct such type of errors.

4)      In Table 3, replace the entry at the first row last column. That should be “Selected”. Correct such types of spelling errors in the whole manuscript.

5)      Only SR and RMSE is not enough. Explore other parameters and include one more parameter.

6)      Before citing an article in the text, put a space. For example, in the 85th line, replace “….which synchronously 85 extracts IMFs from multichannel measurement data[24].” With “…which synchronously 85 extracts IMFs from multichannel measurement data [24].”. Here a space is necessary before [24]. Similar errors exist throughout the manuscript. Correct all such mistakes.

7)      In Introduction, mention your contribution in bullet points. The motivation is also not properly addressed. Rewrite it in the Introduction.

8)      Put a full stop ‘.’ After each figure caption. 

9)      Mention the advantages of PSO in the Introduction.

10)The paper requires extensive modification in the English writing. There are many space, punctuation, grammatical errors. Some sentences are difficult to follow. Rewrite the whole manuscript by correcting the mistakes accordingly

Rewrite the whole manuscript again by correcting grammatical, punctuation, spacing, spelling, and format errors.

Reviewer 3 Report

This manuscript proposes a method for denoising Radio-echo sounding (RES) data using Particle Swarm Optimization and Multivariate Variational Mode De-composition to select the best combination of parameters. MVMD decompose signals in the frequency domain and PSO realize the optimal automated search for the parameters. The reviewer’s comments are as follow. 

1) PSO has a weak local search ability, which means that it may not be able to fine-tune the solutions and escape from local optima. Moreover, PSO may be sensitive to the choice of parameters, such as the number of particles, the inertia weight, the cognitive and social coefficients, and the neighborhood topology. Also, PSO may not be suitable for dynamic or noisy environments, where the objective function or the constraints change over time or are affected by random factors. Please explain about these issues in your research.

2) In abstract please highlight main reason of using Particle Swarm Optimization and Multivariate Variational Mode Decomposition simultaneously.    

3) The introduction part lacks of motivation and organization. The cause and motivation of your method should be inserted in introduction part.

4) Please separate introduction Section into Introduction and Related works.

5) Please indicate the difficulties of denoising Radio-echo sounding (RES) data in introduction. These points should be mentioned in introduction.

6) In introduction, what is the gap? Please specify the gap that your work addresses.

7) In introduction, please insert your new contributions with bullet point.

8) Please indicate about important factors for denoising Radio-echo sounding (RES) data such as the frequency and power of the transmitted radio waves, the dielectric properties of the ice, the noise sources, the filtering methods

9) In related works, please indicate denoising techniques such as low-pass filtering, median filtering, wavelet transform, Fourier transform, sparse representation, low-rank approximation, and deep learning methods and the spaces for RES denoising research.

10) Please show the general framework of proposed method with the input and output data

11) Please provide the pseudo code of the proposed algorithm

12) Please highlight and discuss the equations that you modify.

13) In the equations provide their symbol size and explain the symbols in the formula.

14) Please provide a subsection for explanations about Radio-echo Sounding Data and each dataset. Please insert the properties of your data such as their size. If it is possible, please insert the website address of datasets.

15) Provide a subsection for parameter settings.

16) The proposed method has not been compared with other approaches. Some these methods would be the techniques cited in introduction Section. Please employ recent published articles for comparison.

17) Since there are different types of noise, please provide experiments with other quantitative criteria such as PSNR, and SSIM

18) Please provide qualitative criteria and discuss about some issues such as the denoised data, artificial features or distortions, the contrast between the layers and the background in the denoised data, signals from the deep layers and the bed, strip noise

19) Please emphasize how your work contributes to the current knowledge in the field and explain why your work is important. You can also mention the impact or improvement of your work in the conclusion Section.

Moderate editing of English language required

Round 2

Reviewer 1 Report

Seems like authors did improve the text on the basis of previous comments. 

All the best in future work.

Author Response

请参阅附件。

Reviewer 2 Report

Most of my comments addressed.

I have one comment. Can you provide some tabular representation or visual representation of the comparison of your method with state-of-the-art methods? Justify. If yes, then please provide. This comment is related to my first comment.

NA

Reviewer 3 Report

The current manuscript has been revised according to the comments. However, there are still some concerns that should be revised.

 The reviewer’s comments are as following.

1)    Please indicate the shape and dimension of data in the pseudo code of the proposed algorithm.

2)    For introducing datasets, please indicate the shape and dimension of data.

3)    SSIM criterion is important for the Radio-echo sounding data denoising because it is a measure of the structural similarity. If it is possible, please insert it.

Moderate editing of English language required
